# Neuropsychological Profile of 25 Brazilian Patients with 22q11.2 Deletion Syndrome: Effects of Clinical and Socioeconomic Variables

**DOI:** 10.3390/genes15050595

**Published:** 2024-05-08

**Authors:** Larissa Salustiano Evangelista Pimenta, Claudia Berlim de Mello, Luciana Mello Di Benedetto, Diogo Cordeiro de Queiroz Soares, Leslie Domenici Kulikowski, Anelisa Gollo Dantas, Maria Isabel Melaragno, Chong Ae Kim

**Affiliations:** 1Genetics Unit, Instituto da Criança, Faculdade de Medicina, Universidade de São Paulo, São Paulo 05403-000, Brazil; dr.diogosoares@gmail.com (D.C.d.Q.S.); lesliekulik@usp.br (L.D.K.); chong.kim@hc.fm.usp.br (C.A.K.); 2Departament of Psychobiology, Universidade Federal de São Paulo, São Paulo 04024-002, Brazil; luciana.benedetto@gmail.com; 3Genetics Division, Department of Morphology and Genetics, Universidade Federal de São Paulo, São Paulo 04023-062, Brazil; anegdantas@gmail.com (A.G.D.); melaragno.morf@epm.br (M.I.M.)

**Keywords:** 22q11.2 deletion syndrome, mental disorders, SES, socio-environmental factors phenotype, neuropsychological disorders

## Abstract

The 22q11.2 deletion syndrome (22q11.2DS) is associated with a heterogeneous neurocognitive phenotype, which includes psychiatric disorders. However, few studies have investigated the influence of socioeconomic variables on intellectual variability. The aim of this study was to investigate the cognitive profile of 25 patients, aged 7 to 32 years, with a typical ≈3 Mb 22q11.2 deletion, considering intellectual, adaptive, and neuropsychological functioning. Univariate linear regression analysis explored the influence of socioeconomic variables on intellectual quotient (IQ) and global adaptive behavior. Associations with relevant clinical conditions such as seizures, recurrent infections, and heart diseases were also considered. Results showed IQ scores ranging from 42 to 104. Communication, executive functions, attention, and visuoconstructive skills were the most impaired in the sample. The study found effects of access to quality education, family socioeconomic status (SES), and caregiver education level on IQ. Conversely, age at diagnosis and language delay were associated with outcomes in adaptive behavior. This characterization may be useful for better understanding the influence of social-environmental factors on the development of patients with 22q11.2 deletion syndrome, as well as for intervention processes aimed at improving their quality of life.

## 1. Introduction

The 22q11.2 deletion syndrome (22q11.2DS), also known as DiGeorge syndrome, is caused by a submicroscopic deletion of the long arm of chromosome 22 (MIM #192430#188400). The estimated minimum prevalence of 22q11.2 deletions was 1 in 2148 (4.7 per 10,000) live births [1]. Most patients have approximately a ≈3 megabase (Mb) deletion, with less frequent occurrences of deletions of 2 Mb or about 1.5 Mb [1,2,3].

The main clinical conditions observed in 22q11.2DS deletion syndrome include congenital heart diseases, immunodeficiency from thymic hypoplasia/aplasia, velopharyngeal dysfunction (with or without cleft palate), and hypocalcemia resulting from hypoparathyroidism [1,2,3,4,5,6]. Patients with 22q11.2DS have a high risk of developing epilepsy [7], reported in 11% of cases [8]. However, convulsive episodes are also associated with hypocalcemia, reported in approximately 50% to 69% of patients; symptoms include seizures, tremors, or tetany [9].

Heterogeneity in intellectual performance has been reported by most studies describing the neurocognitive phenotype of 22q11.2DS [10,11]. The full intellectual quotient (IQ) typically ranges from 50 to 109 [10,11,12,13]. A borderline classification (IQ 70–85) is present in more than 50% of cases. Mild and moderate intellectual deficits (IQ 55 to 70) are present in 40% of cases [13,14], and severe levels (IQ < 35) are rare [14]. Only a small percentage of children and adolescents (approximately 15%) show average scores (IQ > 70) [10,11,12]. A significant discrepancy (three to eight points on the standard scale) between verbal and performance IQs is reported in almost 75% of individuals, with better performance in the verbal domain [10,11,15,16,17]. There is evidence of a decline in verbal IQ scores with age, possibly associated with the emergence of psychiatric symptoms [15,16,18].

Neuropsychological findings include deficits in a wide range of cognitive functions, such as attention, executive functioning, and visuospatial processing [11,17,19,20,21,22,23]. Correlates of cognitive and intellectual deficits have been attributed to volumetric alterations in patients with 22q11.2DS [24,25,26,27], as well as to the emergence of psychotic symptoms [18,28,29].

Descriptions of the neurocognitive profile of patients with 22q11.2DS usually focus on psychiatric symptoms [29]. This focus may be attributed to the higher incidence of psychiatric disorders in individuals with this syndrome compared to the general population, as well as to evidence of associations of genomic abnormalities identified by genetic molecular techniques [30]. The risk of schizophrenia has attracted particular interest since [28,29,30,31] it is reported in one-third of patients [31]. Anxiety disorder, depression, and bipolar mood disorder have also been reported [29]. Attention deficit hyperactivity disorder (ADHD) and autism spectrum disorder (ASD) symptoms were most prevalent among children, while the rates of psychoses and mood disorders increased significantly during adolescence and adulthood [11,32,33,34,35].

Neuropsychological and neuropsychiatric characteristics have mainly been attributed to the severity of genetic and neurological conditions. However, the influence of social-environmental variables on phenotypic variability related to cognitive and behavioral skills has not yet been fully explored. Few studies have analyzed the effects of variables such as family socioeconomic status (SES) and caregivers’ schooling [36,37,38,39]. This cross-sectional and exploratory study investigated the neuropsychological profile of a sample of 25 patients with 22q11.2DS and associations among clinical, socioeconomic, and intellectual variables. The results may contribute to a better understanding of the neuropsychological phenotype in 22q11.2 deletion syndrome, as well as to the development of follow-up and intervention planning.

## 2. Materials and Methods

### 2.1. Participants

A sample of 25 patients with 22q11.2DS, of both sexes and aged between 7 and 32 years, participated in the study. All participants were recruited from reference clinical genetic centers affiliated with two Brazilian universities in Sao Paulo City. The diagnosis of 22q11.2DS was clinically established by a medical geneticist and confirmed by molecular investigation using the SALSA MLPA P250-B2 Di-George kit (MRC-Holland, Amsterdam, The Netherlands). All patients exhibited the typical ≈3 Mb deletion. None had been diagnosed with specific psychiatric disorders, including psychosis, at the time of assessment. Information related to access to healthcare indicated that all patients regularly attended medical check-ups at referral clinics and underwent multidisciplinary therapeutic follow-up, such as with psychologists, speech therapists, or occupational therapists.

The ethical committee of the Universidade de Sao Paulo approved this study (process 1088/43/12). Patients and their primary caregivers signed consent forms before the neuropsychological assessment. The entire examination (interviews and testing) was conducted by the main researcher, who is a clinical neuropsychologist.

### 2.2. Procedures

#### 2.2.1. Social-Environmental and Clinical Variables

Social-environmental factors (caregiver years of schooling, access to quality education, literacy, multidisciplinary support), socioeconomic status (family SES), and clinical information (age at diagnosis, pregnancy and birth complications, health conditions, developmental delay, heart defects, and medication use) were obtained through interviews with the main caregivers, typically the mother.

Family socioeconomic status (SES) was categorized based on the household’s gross monthly income and the level of education of the family head, using the Brazilian Economic Classification Criteria [40]. Social classes ranged from B to C, corresponding to monthly incomes equivalent to USD 400 to 1000. Additionally, we considered the type of school the participant attended or had attended. Including the type of school (public or private) as a SES variable was motivated by the significant disparity in the quality of public education in Brazil.

Clinical variables were selected from the diagnostic checklist used by the medical genetic centers where the participants were recruited, in accordance with international guidelines [4,5].

#### 2.2.2. Intellectual, Neuropsychological, and Behavioral Assessment

Intellectual performance was investigated by means of the Brazilian versions of the Wechsler’s Scales for Children (WISC-IV) [41] and adults (WAIS-III) [42], and of the Vineland Adaptive Behavior Scale (VABS-II) [43], answered by the main caretakers. All main indices (standard-scores) and subtests (scaled-scores) of the Wechsler’s scales were considered, as well as VABS main scores (standard-scores): Communication, Daily Living Skills, and Socialization. 

The neuropsychological battery included only cognitive tasks that could be used independently of participants’ ages. The tests’ raw scores were transformed in z-scores and then in t-scores and analyzed descriptively. We used Brazilian norms whenever available. 

Attentional functioning. Continuous Performance Test, 3rd edition—CPT-III [44]. The CPT-III is a computerized procedure widely used for the identification of attention-related problems and is considered as an important additional test for the diagnosis of attention deficit hyperactivity disorder (ADHD). It is used for individuals over 8 years of age. Scores included omissions, commissions, hit reaction time (HRT), and adaptation of reaction time differences in inter stimuli intervals (HIT RT ISI). These scores allow for the inferring of the presence of inattentiveness, impulsivity, deficits of sustained attention, or low vigilance. 

Visuoconstructive skills and visual memory. The Copy of the Complex Figure of Rey was used for the assessment of visuoconstructive skills, as well as for visual long-term memory [45]. Number of elements correctly copied or recalled measured performance. 

Long-term verbal memory. The Rey Auditory Verbal Learning Test (RAVLT) is one of the most frequently used tests in neuropsychological clinical practice [46]. Scores concerning serial recall, recall after interference, and delayed recall were considered for patients’ performance assessment. 

Executive functions. The assessment of executive functioning was based on the inhibitory control and flexibility indexes of the Five Digits Test [47].

### 2.3. Data Analysis

Descriptive statistics were conducted for sociodemographic and clinical characterization of the sample. Exploratory linear regression models were tested in order to investigate which variables would be related to IQ and to the global score of Adaptive Behavior of the VABS-II. In these analyses, t-scores instead of standard-scores were used for comparison purposes. The adequacy of the models was evaluated by the normality of the residues observed in qqplot plots. All analyses were performed in RStudio, considering a significance level of 5%.

## 3. Results

The sociodemographic and clinical characteristics of the sample are presented in Table 1. 

There was a high level of heterogeneity in ages, ranging from 7 to 32 years (22 pediatric participants and 3 adult participants), with a higher frequency of males (64%). Only three participants had a deletion inherited maternally; however, they were raised by grandparents or uncles.

Regarding the participants’ educational characteristics, the vast majority had less than 9 years of schooling. All families reported global academic problems, and 40% of the participants had received therapeutic or pedagogical support at least once in their lives. Only one participant did not attend school due to parental choice and a need for individualized attention.

Clinical information was obtained through interviews with families as well as from medical records. The diagnostic process, including molecular investigation, took approximately 10 years. Cardiac malformations, such as tetralogy of Fallot, interventricular communication defects, aortic stenosis, and truncus arteriosus, were present in 72% of the sample. Some patients underwent surgical interventions to correct these congenital heart defects, which were typically identified at birth. Other clinical problems were reported by caregivers. The most common issues in the first years of life included feeding problems, episodes of otitis media or pneumonia, seizures, and hypocalcemia. The majority of patients (68%) experienced neurodevelopmental delay, especially concerning language milestones.

Parents and adult participants reported several behavioral problems, such as anxiety, depressive mood, socialization difficulties, and impairments in emotional modulation. Long-term pharmacological treatment was reported in these cases, with the most common medications being fluoxetine, methylphenidate, risperidone, clonazepam, and chlorpromazine hydrochloride.

Out of the twenty-five participants recruited for the study, two 10-year-old boys exhibited severe cognitive impairments that hindered formal testing and were subsequently excluded from the sample. Therefore, 23 participants (14 male) underwent the full neuropsychological assessment. Table 2 presents the major intellectual indexes obtained using the WISC-IV and WAIS-III scales.

IQ scores ranged from average (maximum 104) to below-average (minimum 42) classification, with the mean falling into the below-average level (72.30 ± 15.1). Discrepancies among the main indexes of the Wechsler scales were more evident in children than in adult participants. On the WISC IV, the mean verbal comprehension index was higher (82.6 ± 18.8) than the remaining index scores. On the other hand, all results on the WAIS-III were classified at the borderline level. In general, performance on verbal tasks was substantially better than on nonverbal tasks. For eight participants (34.8%), all index scores were compatible with intellectual deficiency (standard scores < 70).

Regarding adaptive abilities, the communication index showed deficits, and all other domains scored in the borderline range (32.0 ± 12.1). Since VABS-II is a questionnaire answered by caregivers, all twenty-five participants were evaluated. The two participants who failed to respond to the intelligence test had the lowest scores.

In the neuropsychological assessment (see Table 3), of the twenty-three patients who underwent intellectual assessment, four did not complete all tasks due to behavioral problems or low adherence to the evaluation procedures. Therefore, some participants only partially completed the battery of tests.

Descriptive results of the whole sample showed average scores on sustained attention tests, indicating that, in general, participants were able to sustain focus for a longer period. Mean reaction times and the ability to adapt responses to changes in the speed of stimulus presentation were as expected based on age. However, the results revealed fluctuations in performance when tasks demanded a higher ability to resist distractions and control automatic responses. In verbal long-term memory tests, average results were observed, and repeated exposure to the material was beneficial.

On the other hand, the poorest results were observed on tasks related to visual short-term memory and visuoconstructive skills, as well as executive functioning. Therefore, nonverbal cognitive functions seem to constitute a cognitive domain of higher vulnerability in 22q11.2DS.

The results of the exploratory linear regression analyses are illustrated in Table 4. 

Several socioeconomic status (SES) variables had significant effects on IQ. Firstly, the type of school had a notable effect (B = 10.00, *p* = 0.0075). It is important to note that in Brazil, there is a significant difference in the quality of education offered by private schools. IQ scores were 10.00 points higher among private school students (mean = 38.00, SD = 10.32) compared to those from public schools (mean = 28.00, SD = 5.94). Additionally, social class had an influence (B = −9.00, *p* = 0.0162): the number of individuals with an IQ score in the average range from classes C or D (mean = 28.00, SD = 6.22) was lower than those from classes A or B (mean = 37.00, SD = 10.23). Finally, there was an effect of caregiver education level, with each one-year increase in education increasing the IQ of 22q11.2DS patients by 1.02 points.

Regarding adaptive abilities, only clinical variables were significantly associated. There was an effect of developmental delay in childhood on global scores. Participants with a history of language delay in childhood had a lower global score (mean = 29.25, SD = 8.50) compared to those who did not (mean = 44.00, SD = 15.13). Additionally, there was an effect of age at diagnosis, with the global score being 0.73 points lower for each year until the diagnosis was made.

## 4. Discussion

The present cross-sectional and exploratory study investigated the neuropsychological profile of a sample of 25 Brazilian individuals diagnosed with 22q11.2DS, all undergoing follow-up in public referral services. We were particularly interested in investigating the extent to which socio-environmental factors (e.g., family income, access to quality education or multidisciplinary intervention) and clinical variables (e.g., age at diagnosis, heart defects) were associated with intellectual outcomes.

Regarding intellectual performance, our sample exhibited a full-scale IQ ranging from 49 to 104 (median 71). Previous studies have also reported IQ scores in the borderline range [10,11]. Only two participants exhibited a restricted cognitive repertoire that prevented formal testing, consistent with previous evidence suggesting that severe intellectual disability (IQ < 35) is uncommon in 22q11.2DS patients [14]. Both of these participants had a clinical history of neonatal complications, suggesting that cognitive impairments may be associated with neurological conditions rather than directly with genotypic-phenotypic associations in 22q11.2DS. In other words, severe intellectual disability could have arisen from secondary clinical problems, such as hypoxic-ischemic events during the neonatal period, cardiac surgeries, brain malformation (polymicrogyria), or neonatal seizures due to hypocalcemia, as reported in the literature [48].

Differences were observed in the intellectual performance of child and adult participants. Among the indexes of the WISC-IV scale, the highest scores were on the verbal comprehension index. A significant discrepancy between the verbal comprehension and perceptual organization indexes has been reported since the first studies on 22q11.2SD [10,12,13,17], particularly in pediatric samples [17]. Studies with larger cohorts of patients have reported declines, particularly in verbal IQ, with increasing age [17,18]. Our results, therefore, reinforce previous evidence that in individuals with 22q11.2DS, verbal reasoning may be better developed in early stages of development, while working memory [49,50], perceptual organization [12,13], and processing speed [51] constitute areas of cognitive weakness.

Regarding adaptive abilities, as reported by parents in the VABS-II scale, significant deficits were mainly detected in the domain of communication. In the remaining domains investigated, daily life activities and socialization, performance was at borderline levels. Adaptive impairments in adolescents and children with 22q11.2DS have been described, usually being worse in the communication domain [34,52]. Lower rates of adaptive behavior have been linked to lower intelligence performance and the presence of psychiatric symptoms [52]. It is also important to consider the role of social cognitive dysfunctions in communication skills. Recently, Jalal et al. (2021) reported in a sample of 50 individuals with 22q11.2SD that deficits in social inference and reciprocal social behavior were more prominent than those shown by individuals with first episode psychosis, clinical high-risk for psychosis, or autism spectrum disorder (mean age = 17.74 ± 5.18; females = 44.3%). Therefore, low language or communicative performance in 22q11.2SD may be explained by social cognitive dysfunctions rather than intellectual deficiency [53].

The neuropsychological profile of our sample of 22q11.2SD patients was characterized by extensive deficits in attentional functioning, including a distracted and impulsive behavioral profile, as well as difficulties in sustaining attention and vigilance. Many studies have described attention problems and an ADHD profile as part of the symptomatology in 22q11.2DS [19,32,54,55]. Significant discrepancies between visuospatial and verbal long-term memory were observed, with verbal memory skills being at average levels. This profile was also reported by previous studies that used similar tasks [56,57]. Worse performance on visual memory tasks may be attributed to visuoconstructive or executive function impairments characteristic of 22q11.2DS, as identified in our sample [11,49,58,59].

Results from the univariate regressions indicated that, among socio-environmental variables, family social class, caretaker educational level, and the patient’s former type of school (private or public) were associated with IQ but not with adaptive behavior. Most studies have shown a stronger association with parental educational level than with family income [37,38,39]. It is possible, then, that in socially disadvantaged populations, higher financial support represents an additional factor in explaining intellectual variability. Additionally, Sashi et al. (2010) demonstrated a role of parental SES in behavioral outcomes in 22q11.2 DS [36].

Regarding the effects of clinical variables, neurodevelopmental delay and age at diagnosis were significantly associated only with adaptive behavior. It is well known that adaptive behavior and IQ, although correlated in typical development, may be dissociated in terms of functional outcomes, as observed in some neurodevelopmental disorders such as autism spectrum disorders [32,60,61]. Dissociations between cognitive skills are related to phenotypic characteristics. In 22q11.2DS, one of the most frequent phenotypic characteristics is language delay [62], as well as language disorders [63]. The implications of language functioning for intellectual development in 22q11.2DS have been investigated. Glaser et al. (2002) compared the performance of samples of children and adolescents with 22q11.2DS and controls with idiopathic developmental delay. The patients showed worse performance in terms of receptive language but not in expressive language when compared with IQ-matched control subjects. Other studies, however, did not identify a direct association between IQ and language deficits [63]. In other words, language dysfunction seems to be a core phenotypic characteristic of 22q11.2SD and therefore is less influenced by intellectual variability. Our results suggest that language may have an impact on overall adaptive behavior development but not on intellectual performance. Selective effects of age at diagnosis on adaptive behavior outcomes were also observed. An early age at diagnosis usually reflects a worse clinical condition. In previous studies, lower rates of adaptive behavior have been linked to low intelligence performance, higher ages, and the existence of psychiatric symptoms [34].

Our study possesses several limitations that restrict the generalizability of the results. The most significant limitation may be the extensive heterogeneity of the sample concerning age and socioeconomic conditions. Therefore, we must acknowledge that our conclusions regarding the effects of socioeconomic and clinical variables on the phenotypic variability in 22q11.2DS may primarily apply to developing countries, where social inequality and limited access to health resources and quality education prevail. Additionally, although the study only included patients without reported psychiatric symptoms, the absence of comprehensive psychiatric evaluations limits the diagnostic confirmation of these findings.

On the other hand, our study has strengths, including the fact that the sample was composed only of individuals with a typical deletion size of ≈3 Mb. It also finds relevance since it contributes to discussions about gene–environment interactions in the 22q11.2SD. For instance, it seems reinforced that the influence of genetic components must be analyzed in line with associated effects of socio-environmental variables, including access to quality of education and to health services. Considering such variables in clinical settings may facilitate a personalization of interventions for intellectual outcomes in this syndrome. 

## 5. Conclusions

Our findings revealed that in a sample of individuals with 22q11.2DS, family income, educational level of the main caretaker, and access to quality education were significantly associated with IQ but not with adaptive behavior, regardless of age at assessment. On the other hand, age at diagnosis and language delay were associated with outcomes in adaptive behavior but not with IQ. These findings indicate that individuals from families living in environments with a greater degree of social vulnerability are at a higher risk for intellectual outcomes. Further studies are necessary for a better understanding of the influence of social-environmental factors on intellectual phenotypic variability in 22q11.2DS.

## Figures and Tables

**Table 1 genes-15-00595-t001:** Sociodemographic and clinical variables of the 22q11.2DS sample (*n* = 25).

**Sociodemographic Variables**	**Min/Max ^1^**	**Mean ± SD**	**Median**
Age at assessment	7/32	13.5 ± 5.0	13.0
Years of schooling	2/11	7.0 ± 2.5	7.0
Caretaker years of schooling	6/24	11.6 ± 4.2	11.0
Gender	*n* (%)		
Male	16 (64)		
	09 (36)		
Type of school			
Public	17 (68)		
Private	08 (32)		
Literate	20 (80)		
Specialized support	10 (40)		
Family social class			
A	4 (16)		
B	6 (24)		
C	10 (40)		
D	5 (20)		
**Clinical Variables**	**Min/Max ^1^**	**Mean ± SD**	**Median**
Age at diagnosis	1/32	9.9 ± 7.2	10.0
	*n* (%)		
Pregnancy problems	6 (24)		
Birth problems	12 (48)		
Poor health conditions	18 (72)		
Language delay	17 (68)		
Heart defects	18 (72)		
Use of psychotropic medications	4 (16)		
De novo deletion	22 (88)		

^1^ Min = minimum; Max = maximum.

**Table 2 genes-15-00595-t002:** Wechsler intelligence scale (*n* = 23) and Adaptive Behavior (*n* = 25) scores observed in the sample of 22q11.2DS deletion syndrome patients.

	Min/Max ^4^	Mean ± SD	Median
FSIQ ^1^	42/104	72.3 ± 15.1	71.0
WISC-IV			
Verbal comprehension	49/119	82.6 ± 18.8	82.0
Perceptual organization	49/104	75.6 ± 14.5	74.0
Working memory	45/109	74.8 ± 20.6	74.0
Processing speed	45/111	72.6 ± 18.2	75.5
WAIS-III			
PIQ ^2^	68/75	71.3 ± 3.5	71.0
VIQ ^3^	72/77	74.7 ± 2.5	75.0
Vineland-II			
Communication	21/103	67.8 ± 18.3	64.0
Daily living skills	38/125	74.0 ± 21.1	68.0
Socialization	42/136	77.1 ± 22.7	69.0
Adaptive Behavior composite	36/126	72.6 ± 19.7	67.0

^1^ FSIQ = full-scale IQ; ^2^ PIQ = performance IQ; ^3^ VIQ = verbal IQ; ^4^ Min = minimum; Max = maximum.

**Table 3 genes-15-00595-t003:** Descriptive data of neuropsychological functioning in the sample of patients with 22q11.2DS (results in t-scores).

	Min-Max	Mean ± SD	Median
Attentional functioning/CPT ^1^ (*n* = 20)			
Omissions	41/90	60.0 ± 13.3	56.0
Commissions	36/84	58.4 ± 12.4	58.0
Hit reaction time (HRT)	32/83	53.5 ± 13.0	52.0
Perseverations	45/90	65.3 ± 16.4	64.0
HRT block change	40/76	56.6 ± 10.9	57.0
HRT ISI change	25/68	46.1 ± 12.1	43.0
Visuoconstructive skills (*n* = 20)			
Copy of the complex Figure of Rey	20/78	28.8 ± 14.3	20.0
Visual short-term memory (*n* = 20)			
Corsi Block backwards	20/58	34.7 ± 11.3	31.5
Verbal episodic memory/RAVLT ^2^ (*n* = 21)			
Serial recall	20/68	42.8 ± 14.7	43.0
Delayed recall	20/63	44.7 ± 13.2	48.0
Recognition (list A)	42/58	41.3 ± 4.4	50.0
Executive function/FDT ^3^ (*n* = 19)			
Inhibition index	20/66	35.2 ± 15.6	31.0
Flexibility index	20/61	36.7 ± 15.7	36.0

^1^ Continuous Performance Test; ^2^ Rey Auditory Verbal Learning Test; ^3^ Five Digits Test.

**Table 4 genes-15-00595-t004:** Regression analysis.

Variables	IQ				Global Adaptive Behavior Scores			
	B	SE	95 CI	*p*	B	SE	95 CI	*p*
Caretaker years of schooling	1.02	0.46	0.062; 1.98	0.0381 *	0.63	0.66	−0.75; 2.01	0.350
Gender ^1^	−4.24	3.92	−12.38; 3.91	0.292	2.76	5.18	−7.99; 13.50	0.600
Type of school ^3^	10.00	3.38	1.33; 17.33	0.0075 *	2.76	5.18	−7.99; 13.50	0.600
Literate ^2^	−9.87	5.42	−21.14; 1.41	0.083	−6.90	6.61	−20.62; 6.82	0.308
Specialized support ^2^	−1.21	4.52	−10.71; 8.29	0.792	6.53	5.32	−4.56; 17.65	0.234
Family social class ^4^	−9.00	3.44	−16.16; −1.84	0.0162 *	−9.09	4.74	−18.92; 0.75	0.0685
Age at diagnosis	−0.31	0.27	−0.87; 0.24	0.256	−0.73	0.33	−1.41; −0.043	0.0383 *
Pregnancy problems ^2^	1.52	5.02	−8.98; 12.03	0.765	5.10	5.95	−7.31; 17.52	0.401
Birth problems ^2^	0.50	4.29	−8.48; 9.48	0.908	6.64	5.19	−4.19; 17.46	0.216
Health conditions ^2^	−5.72	4.86	−15.89; 4.44	0.253	7.96	6.19	−4.94; 20.87	0.213
Language delay ^2^	5.73	4.39	−3.45; 14.91	0.207	14.75	5.05	4.21; 25.29	0.0085 *
Heart defects ^2^	−0.89	4.88	−11.09; 9.29	0.857	3.94	6.25	−9.06; 16.95	0.535
Use of psychotropic medications ^2^	−2.83	5.15	−13.3; 7.87	0.588	6.00	6.65	−7.80; 19.80	0.377

^1^ Female; ^2^ No; ^3^ Private; ^4^ C or D; * *p* < 0.05.

## Data Availability

The data supporting this study’s findings are available from the corresponding author upon reasonable request.

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
