# Peer review of "Neuropsychological Profile of 25 Brazilian Patients with 22q11.2 Deletion Syndrome: Effects of Clinical and Socioeconomic Variables"

_genes, 2024, doi:10.3390/genes15050595_

Round 1
Reviewer 1 Report
Comments and Suggestions for Authors
This study sought to investigate influence of socioeconomic variables on intellectual and adaptive variability in subjects with 22q11.2DS. Taking into account socioeconomic variables is important in clinical practice. The authors report mainly effects of socioeconomic variables on IQ. Clinical variables were predictive of adaptive behavior.
Although an interesting and important topic, the study has some limitations. For starters, the sample is small (N=23 – 25) and consists of both children and adults. This makes the findings difficult to understand as cognitive decline has often been observed in patients with 22q11DS. Therefore, young children and adults may not be very comparable. Moreover, no information about presences of psychiatric disorders is given. Several psychiatric disorders, in particular psychosis are associated with specific neurocognitive impairments and decline. Therefore, presences of psychiatric disorders could have significantly influenced the current findings.
Below my questions and suggestions:
Abstract
-line 19 – 20. Sentences seems incorrect.
Introduction
-A approximately… must be an approximately (line 34)
-The authors state in line 43 that the intelligence quotient varies between 50 and 109. I assume that they refer to a full scale intelligence quotient since the separate indexes can be highly disharmonious.
-The authors state that 20% of patients develops schizophrenia. Other studies, including large scale international studies have reported that 40% of adults developed a psychotic disorder (including schizophrenia). See Schneider et al. 2014. I would recommend adjusting this to 20 – 40%.
Methods
-The authors state that they assessed clinical and sociodemographic variable through interviews. Please provide more information about this.
· Were all the interviews conducted by the same assessor?
· Were predetermined guidelines used to classify this information?
· How was social class determined? What do classes A, B, C, D represent? Please provide a description.
· Did the authors take access to health care into account?
-Why did the authors choose to use T-scores in the Vineland analyses instead of standardized scores typically uses in studies?
-Please specify which scores were used for the neuropsychological outcomes. Are raw scores presented? Or are the scores corrected for age. The latter would be necessary given the large age range in the sample. For example, executive functions develop until ~25 years. Therefore, the raw score of 7 year old is not comparable to that of a 32 year old.
-Do the authors have information on whether or not the deletion was de novo or inherited from one of the parents? This could have a large impact on the sociodemographic status, and parents who also carry the deletion may have more difficulties finding proper health care for their children.
Results
-Please mention how many participants were children and how many were adults.
-In table 3 please also add the test so the reader knows which test corresponds to the outcome.
-Table 4 says QI, this should be IQ? Also, all significant predictors are in bold, accept type of school.
-Did the authors also compare children with adults. As the authors mention, cognitive decline is often observed in 22q11DS.
-It would be interesting the examine the effects of socioeconomic demographics on the separate subscales of the WISC/WAIS. In particular a decline in verbal comprehension is observed. This scale is considered a measure of the acquired knowledge for example at school or from the environment (e.g vocabulary, general information) and may therefore depend more on socioeconomic demographics.
-Considering previously described cognitive decline, in my opinion age of assessment should be included in the regression model, especially given the very large age range in the sample. Moreover, it could be hypothesized that adaptive behavior increases with age as patient may acquire more skills over time.
Discussion
-Can the authors comment on the generalizability of the findings as they may strongly depend on access to specialized care and education which differs between countries.
-The authors should also address ascertainment bias in the limitations. The participants are all recruited through two genetics centers. Possibly, patients in low socioeconomic classes don’t get in contact with health care as easily.
Comments on the Quality of English LanguageSome sentences don't flow very well and need editing.
Author Response
GENES. Special edition
ID: genes-2902348
Paper: Neuropsychological profile of 25 Brazilian patients with 22q11.2 deletion syndrome: effects of clinical and socioeconomic variables
Responses to reviewers.
The authors express sincere gratitude for the reviewers' insightful considerations. Each point raised has been meticulously addressed in the following responses. All modifications have been clearly marked in red within the revised manuscript. Additionally, new references have been thoughtfully incorporated as suggested.
Reviewer 1
Abstract -line 19 – 20. Sentences seem incorrect.
R: We agreed and made changes.
Introduction
-A approximately… must be an approximately (line 34)
R: We corrected the sentence.
-The authors state in line 43 that the intelligence quotient varies between 50 and 109. I assume that they refer to a full-scale intelligence quotient since the separate indexes can be highly disharmonious.
R: Yes, we in fact refer to the full scale. We made changes in the manuscript as follows: The full-scale intellectual quotient (FSIQ) usually varies from 50 to 109 [5,6,7]. (see lines 49-50).
-The authors state that 20% of patients develops schizophrenia. Other studies, including large scale international studies have reported that 40% of adults developed a psychotic disorder (including schizophrenia). See Schneider et al. 2014. I would recommend adjusting this to 20 – 40%.
R: Thank you for your remarks. We made changes in the manuscript and included the paper from Schneider et al. (2014), as follows: Risk of schizophrenia has attracted a particular interest since it is reported one-third of patients [31]. (see line 69).
No information about presences of psychiatric disorders is given. Several psychiatric disorders, in particular psychosis are associated with specific neurocognitive impairments and decline. Therefore, presences of psychiatric disorders could have significantly influenced the current findings.
- Reviewer is absolutely right. Actually, none of our participants presented psychiatric symptoms according to medical reports at the time of the assessment. We included a sentence in the manuscript in this regard: None had been diagnosed with specific psychiatric disorders, including psychosis, at the time of assessment. Information related to access to health care indicated that all patients regularly attended medical check-ups at referral clinics and underwent multidisciplinary therapeutic follow-up, such as with psychologists, speech therapists, or occupational therapists. (see lines 92-97).
Methods
The authors state that they assessed clinical and sociodemographic variable through interviews. Please provide more information about this.
- Were all the interviews conducted by the same assessor?
R: Yes. We included the following sentence in lines 100-101: The entire examination (interviews and testing) was conducted by the main researcher, who is a clinical neuropsychologist.
- Were predetermined guidelines used to classify this information?
R: Both clinical and sociodemographic information were obtained by medical records and checked in the interviews with the main caretaker. Regarding the socio demographic variables, we considered among others family income, according to the Brazilian Economic Classification Criteria. We considered the main clinical variables from the diagnostic checklist adopted by the clinical genetic centers where the participants were recruited. A full paragraph was inserted in the manuscript (see lines 105-119).
- How was social class determined? What do classes A, B, C, D represent? Please provide a description.
R: We included a sentence in lines 105-119 in the above mentioned paragraph: This means that monthly income in “reais”, the official currency of Brazil, varied equivalent to USD 400 to 1000 and most of the primary caregivers completed high school or college. Thank you for this observation, we hope that this information has become clearer.
- Did the authors take access to health care into account?
R: Thank you so much for this observation. The answer is yes, but we indeed did not include such information in the manuscript. We included the following sentence in the manuscript: Information related to access to health care indicated that all patients regularly attended medical check-ups at referral clinics and underwent multidisciplinary therapeutic follow-up, such as with psychologists, speech therapists, or occupational therapists. (line:94-97).
- Why did the authors choose to use T-scores in the Vineland analyses instead of standardized scores typically used in studies?
R: We converted the Vineland´s standard-scores into t-scores for comparison purposes. We included this information in the methods. Thank you. (see lines 154-155).
-Please specify which scores were used for the neuropsychological outcomes. Are raw scores presented? Or are the scores corrected for age. The latter would be necessary given the large age range in the sample. For example, executive functions develop until ~25 years. Therefore, the raw score of 7 year old is not comparable to that of a 32 year old.
R: We converted the Vineland´s standard-scores into t-scores for comparison purposes. We included this information in a sentence in line 154 (Data Analysis section). Thank you.
- Do the authors have information on whether or not the deletion was de novo or inherited from one of the parents? This could have a large impact on the sociodemographic status, and parents who also carry the deletion may have more difficulties finding proper health care for their children.
R: In our sample there were only three cases of inherited deletion, and all from their mothers. They were not raised by their biological parents. Their main caretakers and legal guardians were relatives. We agreed that this in an important question and included data regarding inheritance patterns in Table 1 and new sentences in the discussion topic (see table 1 and lines 168-169).
Results
Please mention how many participants were children and how many were adults.
R: We included this relevant information in the following sentence: There was a high level of heterogeneity in ages, ranging from seven to 32 years (22 children and 03 adults), with a higher frequency of males (64%). (line 166-167).
- In table 3 please also add the test so the reader knows which test corresponds to the outcome.
R: Corrected. Thank you for the careful reading.
- Table 4 says QI, this should be IQ? Also, all significant predictors are in bold, accept type of school.
R: Corrected. Thank you for the careful reading.
- Did the authors also compare children with adults? As the authors mention, cognitive decline is often observed in 22q11DS.
R: We considered that due to the small number of adult patients a comparison by age was not appropriate. Additionally our study followed a more descriptive and exploratory design with a focus on the influences of socio-environmental variables. But since this is a relevant issue, we decided to include a sentence in final of the topic discussion for future analysis (see lines 337-345)
- It would be interesting the examine the effects of socioeconomic demographics on the separate subscales of the WISC/WAIS. In particular a decline in verbal comprehension is observed. This scale is considered a measure of the acquired knowledge for example at school or from the environment (e.g vocabulary, general information) and may therefore depend more on socioeconomic demographics.
R: We absolutely agree that this is a relevant issue. However we understood that the discrepancy between the adults and children sample sizes would made analysis difficult. Nevertheless, we also included some sentences in the discussion as themes for future analysis (see lines 337-345).
- Considering previously described cognitive decline, in my opinion age of assessment should be included in the regression model, especially given the very large age range in the sample. Moreover, it could be hypothesized that adaptive behavior increases with age as patient may acquire more skills over time.
R: Discussing the issue of decline was beyond the scope of our study, which focused on a heterogeneous population seeking services at a high-demand referral center specialized in 22q11.2 deletion syndrome in Brazil. A descriptive analysis that focuses on the potential influence of adaptive and intellectual domains of intelligence already addresses an important question for public policy interventions. However, we believe that future studies on this topic would be crucial for addressing new and relevant questions in the field of this syndrome.
Discussion
Can the authors comment on the generalizability of the findings as they may strongly depend on access to specialized care and education which differs between countries.
R: That is a very important part of the discussion, thank you. We included a paragraph in this regard in lines 337-343:
The authors should also address ascertainment bias in the limitations. The participants are all recruited through two genetics centers. Possibly, patients in low socioeconomic classes don’t get in contact with health care as easily.
R: That is a very important part of the discussion, thank you. We included a paragraph in this regard in lines 337-350:
Comments on the Quality of English Language. Some sentences don't flow very well and need editing.
R: We will provide a full english revision. Thank you.

Reviewer 2 Report
Comments and Suggestions for Authors
22q11.2DS is known to be associated with an heterogeneous neurocognitive phenotype, including psychiatric disorders being some of them age-dependent. Neuropsychological and neuropsychiatric characteristics have been attributed mainly to the severity of genetic and neurological conditions. However, The influence of social-environmental variables on phenotypic variability related to cognitive and behavioral skills have not yet been fully explored. The apparent aim of the present study was to assess the hypothesis that socioeconomic variables could influence intellectual variability/ cognitive phenotype in22q11.2DS, explaining its wide neurocognitive heterogeneity. The study sample consists in n=25 patients (age: 7 to 32 years) all harboring a typical ~3 Mb 22q11.2 deletion. Cognitive performance was assessed by WISC-IV, WAIS-III, and VABS-II scales administered to caregivers; CPT-III was used to test attention; visuoconstructive skills and visual Memory were tested by the Copy of the Complex Figure of Rey; Long-term Verbal Memory was tested by RAVLT. SES was assessed through family income and quality schooling data. Statistics is univariate. IQ ranged widely (42 to 104). Communication, executive functions, attention and visuoconstructive skills were found to be mostly impaired in the examined population. From the results, the AA conclude that type of school, social class and caregiver education have an effect on IQ, language delay and global adaptive behavior in the 22q11.2DS.
Main Points
1 -Study Limitations: in their discussion, the AA already expose a number of potential bias factors. Using a simple univariate statistical approach is another potential limitation to be added.
2. Some of the Methods are mistakenly described in the Results section (lines 145-152).
3. - Discussion is rather confusing and dysorganized. The AA should reorganize and improve the conceptual architecture of the Discussion to better convey their message.
4. -The study conclusion is, my own view, unconvincing as the AA could not prove a cause-effect relationship but rather show statistical associations. This point should be adequately addressed.
5. The cited Literature includes a majority of old citations given that only 5/39 of the references are from the last 5 years (2020-2024, i.e., apx 13% of the total Refs list). Most importantly, at least one reference (Ge et al, 2024), potentially relevant to the present research, is not cited nor discussed by the AA. Therefore, I would encourage the AA to do so.
Suggested Reference:
Ge R, et al. Source-based morphometry reveals structural brain pattern abnormalities in 22q11.2 deletion syndrome. Hum Brain Mapp. 2024 Jan;45(1):e26553. doi: 10.1002/hbm.26553.
6. -The AA state that their “Results may contribute to better understanding of neurocognitive phenotype in the 22q11.2 deletion syndrome, as well as to developmental follow-up and planning of interventions”. However, the AA should better explain how their results will be able to do this.
Comments on the Quality of English Language
Language needs extensive revision.
Author Response
GENES. Special edition
ID: genes-2902348
Paper: Neuropsychological profile of 25 Brazilian patients with 22q11.2 deletion syndrome: effects of clinical and socioeconomic variables
Responses to reviewers.
The authors express sincere gratitude for the reviewers' insightful considerations. Each point raised has been meticulously addressed in the following responses. All modifications have been clearly marked in red within the revised manuscript. Additionally, new references have been thoughtfully incorporated as suggested.
Reviewer 2
Thank you for your relevant considerations. We tried to answer each one, as follows. All changes are marked in red in the revised manuscript.
- Study Limitations: in their discussion, the AA already expose a number of potential bias factors. Using a simple univariate statistical approach is another potential limitation to be added.
- That is a very important part of the discussion, thank you. We included a paragraph in this regard in lines 337-350:
- Some of the Methods are mistakenly described in the Results section (lines 145-152).
R: Corrected. Thank you for the careful reading.
- Discussion is rather confusing and dysorganized. The AA should reorganize and improve the conceptual architecture of the Discussion to better convey their message.
R: We have agreed to the suggested changes and have accordingly revised portions of the text. New references have been thoughtfully incorporated. Thank you!
- The study conclusion is, my own view, unconvincing as the AA could not prove a cause-effect relationship but rather show statistical associations. This point should be adequately addressed.
R: That is a very important part of the discussion, thank you. We included a paragraph in this regard.
“Our findings revealed that in a sample of individuals with 22q11.2DS, family income, educational level of the main caretaker, and access to quality education were significantly associated with IQ but not with adaptive behavior, regardless of age at assessment. On the other hand, age at diagnosis and language delay were associated with outcomes in adaptive behavior but not with IQ…” (see lines 354-365)
- The cited Literature includes a majority of old citations given that only 5/39 of the references are from the last 5 years (2020-2024, i.e., apx 13% of the total Refs list). Most importantly, at least one reference (Ge et al, 2024), potentially relevant to the present research, is not cited nor discussed by the AA. Therefore, I would encourage the AA to do so.
Suggested Reference:
Ge R, et al. Source-based morphometry reveals structural brain pattern abnormalities in 22q11.2 deletion syndrome. Hum Brain Mapp. 2024 Jan;45(1):e26553. doi: 10.1002/hbm.26553.
R: We totally agreed and made several changes
- The AA state that their “Results may contribute to better understanding of neurocognitive phenotype in the 22q11.2 deletion syndrome, as well as to developmental follow-up and planning of interventions”. However, the AA should better explain how their results will be able to do this.
R: We totally agreed. We included the following sentence.
By demonstrating that variables related to socioeconomic background can influence intellectual outcomes, our results contribute to a better understanding of the variability of neurocognitive phenotype in the 22q11.2SD and highlight the potential effects of early cognitive stimulation. We included a paragraph in this regard. (see lines 354-365).
- Comments on the Quality of English Language
Language needs extensive revision.
R: We submitted the full text to extensive English revision

Round 2
Reviewer 1 Report
Comments and Suggestions for Authors
The authors have addressed all my questions. I have no other suggestions.
Comments on the Quality of English LanguageThe use of English language has improved.
Reviewer 2 Report
Comments and Suggestions for Authors
To The AA
The AA have carefully addressed each of the raised criticism points. The Ms is significantly improved.
Comments on the Quality of English LanguageThe Ms is still in need of minor text editing that could be easily done during the proof reading phase.